# Comparison of Yamuna (India) and Mississippi River (United States of America) bacterial communities reveals greater diversity below the Yamunotri Glacier

Osvaldo Martinez[1]*, Silas R. Bergen[2], Jacob B. Gareis[2]

1 Biology Department, Winona State University, Winona, MN, United States of America, 2 Mathematics and Statistics Department, Winona State University, Winona, MN, United States of America

* omartinez@winona.edu

**Data Availability Statement:** The sequence data was uploaded into the NCBI SRA database. The BioProject accession number is PRJNA1077596.

## Abstract

The Yamuna River in India and the Mississippi River in the United States hold significant commercial, cultural, and ecological importance. This preliminary survey compares the bacterial communities sampled in surface waters at 11 sites (Yamuna headwaters, Mississippi headwaters, Yamuna River Yamunotri Town, Mississippi River at Winona, Tons River, Yamuna River at Paonta Sahib, Yamuna River Delhi-1, Yamuna River Delhi-2, Yamuna River before Sangam, Sangam, Ganga River before Sangam). Bacterial 16S rDNA analyses demonstrate dominance of Proteobacteria and Bacteroidetes phyla. Actinobacteria were also dominant at sites near Sangam in India and sites in Minnesota. A dominance of Epsilonbacteraeota were found in Delhi, India. Principal component analysis (PCA) using unique operational taxonomic units (OTUs) resulted in the identification of 3 groups that included the Yamuna River locations in Delhi (Delhi locations), Yamuna headwaters and Yamuna River at Yamunotri (Yamuna River locations below the Glacier) and Mississippi, Ganga, Tons, and other Yamuna River locations. Diversity indices were significantly higher at the Yamuna River locations below the Glacier (Simpson $D = 0.986$ and Shannon $H = 5.06$) as compared (p value <0.001) to the Delhi locations ($D = 0.951$ and $H = 4.23$) and as compared (p value < 0.001) to Mississippi, Ganga, Tons, and other Yamuna River locations ($D = 0.943$ and $H = 3.96$). To our knowledge, this is the first survey to compare Mississippi and Yamuna River bacterial communities. We demonstrate higher diversity in the bacterial communities below the Yamunotri glacier in India.

## Introduction

The Mississippi and Yamuna rivers play important roles in the cultures, ecologies, and economies of the USA [1] and India [2], respectively. Whereas the source of the Yamuna River is the Yamunotri Glacier located in the Himalayan mountains of the state of Uttarakhand, the headwaters of the Mississippi River are at Lake Itasca located in the center of the northern

**Funding:** OM is a recipient of personal improvement fund (PIF), Winona State University Faculty-led Program Exploratory and Winona Foundation Grants. JG was funded through the Winona State University Work Study Program. The funders had no role in study design, data collection and analysis, decision to publish, or preparation of the manuscript.

**Competing interests:** The authors have declared that no competing interests exist.

Midwestern state of Minnesota (MN). The Mississippi River basin drains more water from the continental United States and Canada than any other North American river. The Yamuna River, largest tributary of the Ganga River, has a basin that drains the southern Himalayas and crosses the states of Uttarakhand and Uttar Pradesh [2].

Both the Mississippi and Yamuna Rivers serve as a source of not only drinking water, but support industrial, recreational and agricultural activities. Anthropogenic pressure on the rivers varies with location, population and industry. For example, in the United States, the upper Mississippi River valley agricultural, forested and urbanized land use varies from 8.7%, 2.2% and 70.8%, respectively at the relatively pristine lake headwaters in Itasca to 9.2%, 77.8%, and 3.8%, respectively at the Twin cities (population estimated to be 3 million in 2024) and 37.7%, 14.8%, and 25.3% respectively at La Crescent, 43.9 km from Winona, MN [3]. In India, at the source of the Yamuna River, within the upper Yamunotri valley in Uttarakhand, only 4% is cropland, 24% is grass land, 14.7% is snow and glaciers, 0.7% deciduous forest, 44.5% is evergreen/semievergreen forest, 9.9% is barren rock, 0.23% is "built up" and only 1.5% is river water [4]. However, approximately 315 kilometers (kms) downstream of the Yamunotri glacier, the Yamuna River faces enormous anthropogenic pressure by a Delhi population of 16.8 million inhabitants that grows 18–90% every ten years as measured by the latest census in 2011 [5]. This pressure is exacerbated by an infrastructure that is inadequate for dealing with an ever-increasing demand for water and its treatment [5–9].

Previous studies surveying the composition of bacterial river communities in the Yamuna and Mississippi rivers have been used to identify sediment and surface water bacteria, as well as the effects of pollution, seasonal changes, temperature, and other parameters on bacterial community composition [3, 7, 9–26]. Riverine bacterial community composition has been identified by sequencing total DNA from harvested river water and sediment and comparing the sequence of ubiquitous and evolutionarily conserved genes such as 16S rDNA that are used to identify bacterial taxa [3, 27]. Further, sequencing of whole bacterial genomes (WGS) has been used characterize potential bacterial functions.

In this study, we used 16S rDNA sequencing of total harvested water DNA to identify and compare the bacterial communities of both the Mississippi and Yamuna river headwaters and 9 downstream sites including the Mississippi River at Winona, Yamuna River at Yamunotri, confluence of the Tons and Yamuna River, Yamuna River at Paonta Sahib, Yamuna River Delhi-1, Yamuna River Delhi-2, Yamuna River before Sangam, Yamuna and Ganga River mixture at Sangam, and Ganga River before Sangam. Our comparison of the Yamuna and Mississippi rivers demonstrated the dominance of Proteobacteria and Bacteroidetes in all assayed locations and the expansion of Epsilonbacteraeota in the Yamuna River in Delhi. Furthermore, we show that bacterial communities closest to the Yamunotri Glacier showed highest Shannon and Simpson diversity indices as compared to bacterial communities at other river locations.

## Materials and methods

### Harvesting microbes

Microbes were isolated in triplicate within 4 hours of harvesting surface (harvested <10 cm deep) river water during late morning and afternoon. River water (50 mL) was gravity-filtered through a 3MM Whatman filter to eliminate large particulate matter. The next step involved filling a 50-milliliter syringe with river water and utilizing its plunger to push the water through pre-cut 0.22-micron PVDF non-sterile hydrophilic filters (GVSP04700, Millipore Sigma, St. Louis, MO) held in a 25 mm re-usable syringe holder (EW-29550-42, Cole-Parmer, Vernon Hills, IL). The microbe-immobilized filter membranes were preserved in DNA/RNA Shield Lysis Tubes (R1103, Zymo Research, Irvine, CA) until DNA could be harvested and processed.

## Harvesting microbe DNA

Microbe DNA was harvested according to manufacturer's protocol using ZymoBIOMICS DNA microPrep Kit (Zymo Research). Briefly, DNA/RNA Shield Lysis Tubes containing bacteria-immobilized filters and bashing beads were vortexed for 15 minutes. Lysis tubes were centrifuged and 400 μL lysate was harvested and transferred into a new tube along with 800 μL ZymoBIOMICS DNA Binding Buffer. Next, 800 μL of this mixture was then transferred to a Zymo-Spin IC-Z Column, spun at 10 000 x g for 1 min and flow through was discarded. The previous step was repeated to harvest DNA from the remaining lysis and DNA Binding Buffer mixture. Column DNA was washed with 400 microliters ZymoBIOMICS DNA Wash Buffer 1, 700 μL ZymoBIOMICS DNA Wash Buffer 2 and 200 μL ZymoBIOMICS DNA Wash Buffer 2. Each wash was centrifuged at 10 000 x g for 1 minute, flow through discarded. The microbe DNA in the Zymo-Spin IC-Z C column was eluted after 1 minute of incubation with 25 μL of ZymoBIOMICS DNase/RNase Free Water. Eluted DNA was further spun through a prepared Zymo-Spin II-micro HRC Filter at 16 000 x g for 3 minutes. DNA was then quantitated and processed for 16S rDNA sequencing.

## DNA processing and bioinformatics

Total DNA samples were sent to Igenbio (Igenbio, Chicago, IL) for library preparation, sequencing, bioinformatics, and bacterial identification. The V3-V4 region of the 16S Ribosomal Gene was amplified using primers specific for those regions (515F: GTGCCAGCM GCCGCGGTAA and 806R: GGACTACHVGGGTWTCTAAT) [28] using a two-step PCR process [29]. Samples were barcoded with linker sequences (Fluidigm Access Array Barcodes). DNA Quality was assessed using an agarose gel electrophoresis. Illumina Libraries were created and sequenced (MiniSeq, Illumina, San Diego, CA). Sequencing yielded approximately 6.1 million paired-end reads of 153bp. Sequence quality was assessed using ERGO's [30] Read Quality Workflow.

Primers were removed from the reads using cutadapt [31]. Amplicon Sequence Variants (ASVs) were derived using ERGO's Amplicon Sequence Workflow which utilizes DADA2 [32]. Reads were denoised and then merged. Afterwards, chimeric sequences were removed retaining 95.5% of sequences. Next, taxonomy was assigned using a naïve bayesian classifier [33] built on version 138.1 of the SILVA 16S database [34] specific to the 515F-806R region. This resulted in 13,466 unique ASVs across 36 samples. The sequence data was uploaded into the NCBI SRA database. The BioProject accession number is PRJNA1077596 containing 16SrDNA from Yamuna River at Paonta Sahib (SRX23651587, SRX23651555, SRX23651554), Yamuna River at Yamunotri (SRX23651586, SRX23651585, SRX23651584), Yamuna headwaters (SRX23651583, SRX23651582, SRX23651575), Mississippi River at Winona (SRX23651581, SRX23651580, SRX23651579), Mississippi headwaters (SRX23651553, SRX23651564, SRX23651552), Ganga and Yamuna Rivers at Sangam (SRX23651572, SRX23651573, SRX23651574), Ganga River before Sangam (SRX23651569, SRX23651570, SRX23651571), Yamuna River before Sangam (SRX23651566, SRX23651567, SRX23651568), Tons River (SRX23651562, SRX23651563, SRX23651565), Yamuna River at Delhi-1 (SRX23651556, SRX23651557, SRX23651558), Yamuna River at Delhi-2 (SRX23651559, SRX23651560, SRX23651561).

## Principal component analysis, Richness and diversity analysis and ANOVA

We carried out principal component analysis (PCA) on the relative abundances of OTUs at each location. PCA was used to identify groups of locations with similar bacterial profiles and explore differences in bacterial communities between groups of locations. Following PCA we

also compared mean species richness (measured by total number of OTUs) and diversity (measured with the Simpson and Shannon diversity indices) of the river location groups as identified by PCA. For $K$ unique OTUs observed at a single location, the Simpson diversity index of that location is given by:

$$D = 1 - \frac{\sum_{i=1}^{K} n_i(n_i - 1)}{N(N - 1)},$$

where $n_i$ represents the number of observed $OTU_i$ and $N = \sum_{i=1}^{K} n_i$ represents the total number of OTUs counted at that location. The Shannon diversity index of a location is given by:

$$H = - \sum_{i=1}^{K} p_i \cdot \ln(p_i),$$

where $p_i = n_i/N$ is the proportion of the total number of observed OTUs at that location made up of $OTU_i$. After computing the richness, Simpson's $D$, and Shannon's $H$ for all samples, the means of these metrics were compared across PCA groups with one-way ANOVA. Groups that demonstrated statistically significant differences were subsequently analyzed with Tukey's honestly significant difference (HSD) post-hoc pairwise comparisons to determine which groups were responsible for those differences.

## Results

In this study, we compared the bacterial communities found in the waters of two rivers, the Upper Yamuna River in India and the Upper Mississippi River in the USA. Surface water was harvested from the Mississippi headwaters at Lake Itasca (Fig 1A) and the Yamuna headwaters in the Himalayas at the Yamunotri Glacier in Uttarakhand, India (Fig 1B and 1C). Other sample locations included the Mississippi River at Winona, MN (Fig 1A), the Yamuna River at Yamunotri (Janki Chatti), Tons River at the confluence of the Yamuna River (Fig 1B and 1C), the Yamuna River at Paonta Sahib (Fig 1B and 1C), two Yamuna River locations in the city of Delhi, approximately 6 meters apart, at the Dayadham Yamuna Aarti Ghat (Fig 1B), the Yamuna River approximately 6.5 kms upstream from Sangam and the Ganga River approximately 3 kms upstream from Sangam and the Yamuna and Ganga River mix at Sangam (Fig 1B). Table 1 shows collection date, water sample temperature and location coordinates for all locations. Harvested water temperatures were measured to be 27, 23.2, 19.3, 18.6, 25.6, 25.3, 32.9, 32.9, 30.9, 32.6, 31.1 at the Mississippi headwaters, Mississippi River at Winona, Yamuna headwaters, Yamuna River at Yamunotri, Tons River, Yamuna River at Paonta Sahib, Yamuna River in Delhi-1, Yamuna River in Delhi-2, Yamuna River upstream of Sangam, Ganga River upstream of Sangam, Yamuna and Ganga River at Sangam, respectively.

Total DNA harvested from filtered Mississippi River and Yamuna River samples was processed for 16S rDNA sequencing and operational taxonomic unit (OTU) identification using a bacterial 16S rRNA sequence database. Regardless of river site, 16S rDNA sequence analysis of triplicate samples demonstrated a dominance of Proteobacteria and Bacteroidetes phyla (Fig 2). Proteobacteria dominated the Mississippi River at the Headwaters (37%) and Winona (36%), as well as the Yamuna River at the Headwaters (49%) and the Yamunotri town (51%) downstream of the Yamunotri Glacier, the Tons River (44%), the Yamuna River at Paonta Sahib (46%), Delhi sites 1 (43%) and 2 (45%), Yamuna River at Sangam (41%), mix of the Yamuna and Ganga Rivers at Sangam (51%) and the Ganga River upstream of Sangam (40%) (Fig 2). The Bacteroidetes phyla also dominated the bacterial communities found at the Mississippi River at the Headwaters (25%) and Winona (22%), as well as the Yamuna River at the

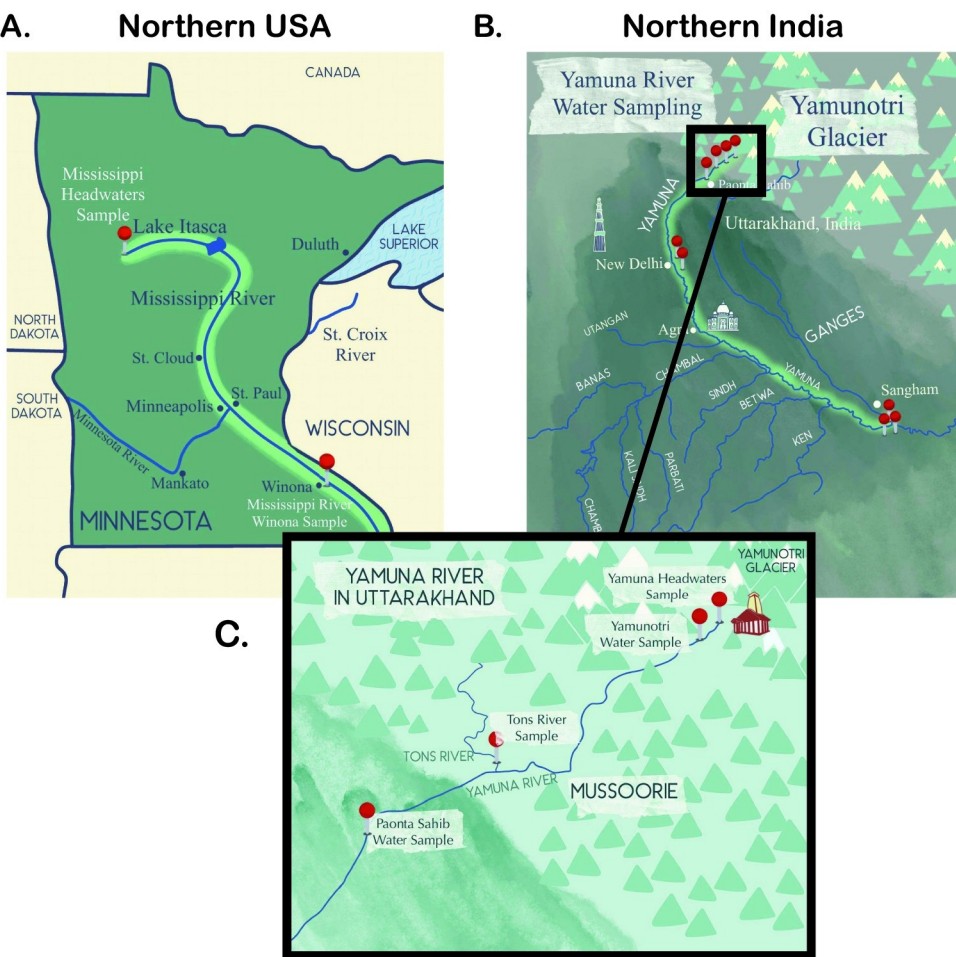

**Fig 1. Sampling locations.** Shown in red pins are the locations surface river water was harvested from (A) Minnesota, USA, (B) state of Uttarakhand, India and (C) is a magnification of the boxed area shown in (B).

**Table 1. Collection date, water sample temperature and location coordinates.**

| Water sample source | Date water collected | Coordinates | Water sample temperature (Celsius) |
|---|---|---|---|
| Mississippi headwaters | 7-13-19 | 47°14'22.8"N 95°12'28.0"W | 27 |
| Mississippi River at Winona | 7-12-19 | 44°03'18.2"N 91°38'07.7"W | 23.2 |
| Yamuna headwaters | 7-20-19 | 31°00'36.5"N 78°27'40.3"E | 19.3 |
| Yamuna River at Yamunotri | 7-20-19 | 30°58'35.3"N 78°26'19.8"E | 18.6 |
| Tons River | 7-21-19 | 30°30'38.0"N 77°49'03.1"E | 25.6 |
| Yamuna River at Paonta Sahib | 7-21-19 | 30°26'04.1"N 77°37'14.9"E | 25.3 |
| Yamuna River in Delhi-1 | 7-22-19 | 28°40'33.8"N 77°13'54.7"E | 32.9 |
| Yamuna River in Delhi-2 | 7-22-19 | 28°40'32.3"N 77°13'55.4"E | 32.9 |
| Yamuna River upstream of Sangam | 7-28-19 | 25°25'41.3"N 81°52'37.7"E | 30.9 |
| Ganga River upstream of Sangam | 8-2-19 | 25°26'22.3"N 81°53'01.2"E | 32.6 |
| Yamuna and Ganga River at Sangam | 7-28-19 | 25°25'36.1"N 81°53'13.8"E | 31.1 |

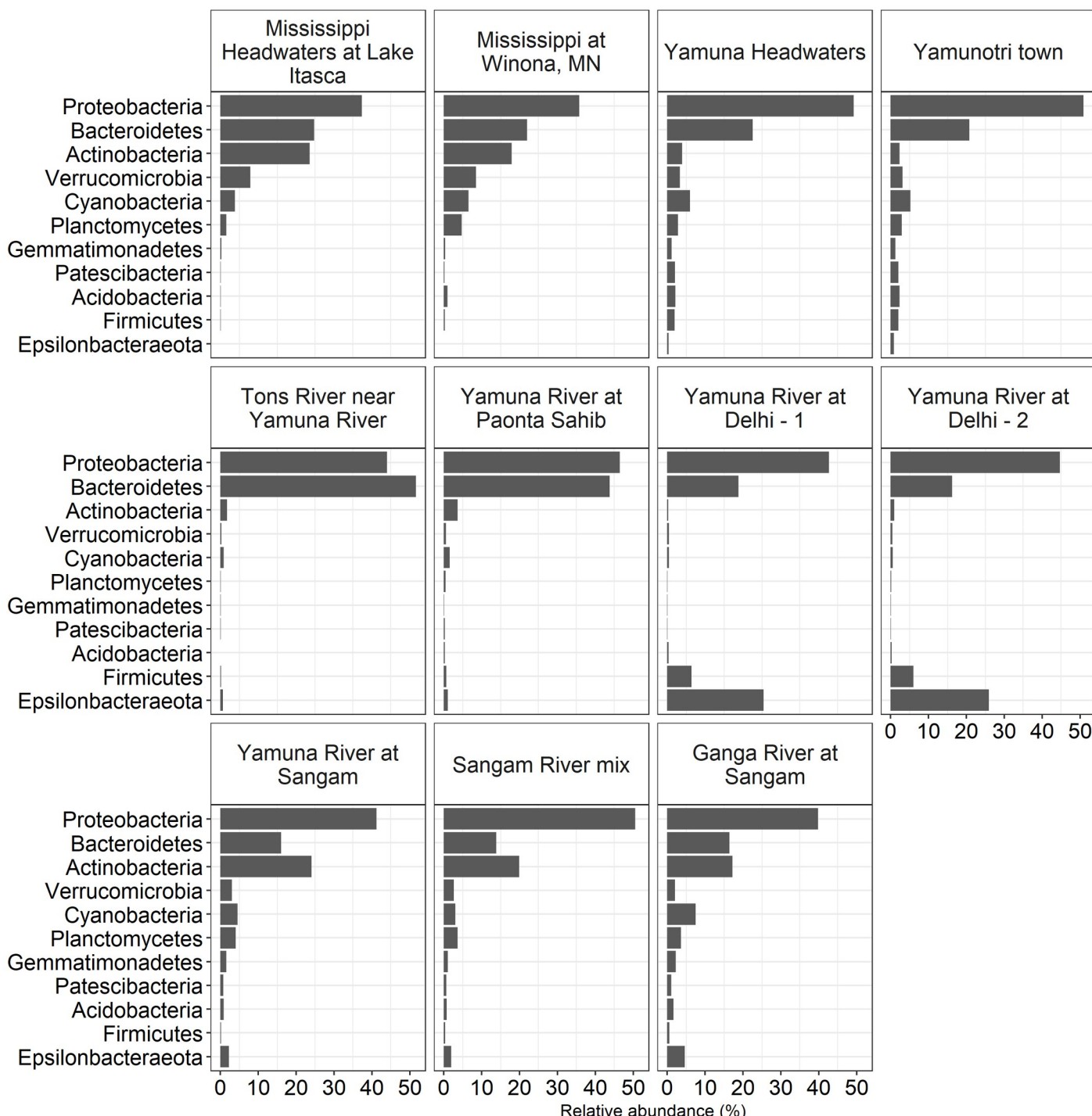

**Fig 2. Most abundant phyla.** Relative abundance of phyla aggregated across triplicate samples at each of the Mississippi headwaters, the Mississippi River at Winona, Yamuna headwaters, the Yamuna River at the town of Yamunotri, the Tons River at its confluence with the Yamuna River, Yamuna River at Paonta Sahib, Yamuna River in Delhi-1, Yamuna River in Delhi-2, Yamuna River upstream of Sangam, Yamuna and Ganga River mixture at Sangam in Allahabad, Ganga River upstream of Sangam locations. Only phyla with at least 2% relative abundance at any location are shown.

Headwaters (23%) and the Yamunotri town (21%) downstream of the Yamunotri Glacier, the Tons River (52%), the Yamuna River at Paonta Sahib (44%), Delhi sites 1 (19%) and 2 (16%), Yamuna River at Sangam (16%), mix of the Yamuna and Ganga Rivers at Sangam (14%) and the Ganga River upstream of Sangam (16%) (Fig 2).

The Actinobacteria were also prominent at the Mississippi River Headwaters (23%) and Winona (18%), but not at the Yamuna Headwaters (<5%) and the Yamunotri town (<5%) downstream of the Yamunotri Glacier, nor at the Tons River (<5%), the Yamuna River at Paonta Sahib (<5%), nor in Delhi sites 1 (<5%) and 2 (<5%). However, Actinobacteria did contribute to the bacteria phyla found at the Yamuna River at Sangam (24%), mix of the Yamuna and Ganga Rivers at Sangam (20%) and the Ganga River upstream of Sangam (17%) (Fig 2). The phyla Epsilonbacteraetoa was prominently represented only in the Yamuna River at Delhi sites 1 (25%) and 2 (26%), but not in any of the other locations.

Figs 3–4 show the most common OTUs at each location. All locations except for the Yamuna headwaters and the Yamuna River at the town of Yamunotri showed that the bacterial communities tended to be dominated by just 2 unique OTUs. The two unique OTUs that dominated the bacterial communities found at the Mississippi River Headwaters and at Winona were the Actinobacteria (phyla); Sporichthyaceae (family); *hgcl_clade* (genus) and the Proteobacteria (phyla); SAR11_clade (family); *Clade_III* (genus) (Fig 3A and 3B). On the other hand, two different bacterial OTUs (Fig 3E and 3F), Bacteroidetes (phyla); Flavobacteriales (family); *Flavobacterium* (genus) and Proteobacteria (phyla); Burkholderiaceae (family) were dominant in the Tons River and Yamuna River at Paonta Sahib. A separate couple of bacterial OTUs (Fig 4A and 4B) dominated the Yamuna River sites in Delhi. These included the Proteo-bacteria (phyla); *Geobacter* (genus) and the Epsilonbacteraeota (phyla); *Arcobacter* (genus). Further downstream, the dominant OTUs (Fig 4C–4E) at all Sangam sites were Actinobacteria (phyla); Ilumatobacteraceae (family); *CL500-29_marine_group* (genus) and Actinobacteria (phyla); Sporichthyaceae (family); *hgcl_clade* (genus).

The biplot in Fig 5 shows the first two principal components from the principal component analysis (PCA). These two components explained a combined 47.1% of the total variability in OTU relative abundance among the 11 locations. PCA appeared to cluster locations with similar bacterial profiles into three groups. The groups consisted of the two Yamuna River locations just below the Yamunotri Glacier; the two Yamuna River locations in Delhi; and the Mississippi River locations and other Yamuna River or adjacent river locations.

We next compared the most abundant bacterial phyla (Fig 6) and OTU (Fig 7) profiles from the PCA Yamuna River below Glacier, Yamuna River at Delhi and Mississippi, Ganga, Tons and other Yamuna River locations groups. Proteobacteria and Bacteroidetes were well-represented in all three groups (Fig 6). The Yamuna River at Delhi locations had a higher abundance of Epsilonbacteraeota and Firmicutes as compared to all the other groups, while the Mississippi, Ganga, Tons, and other Yamuna River locations had a higher abundance of Actinobacteria (Fig 6). With respect to the distribution of top 10 OTUs (Fig 7), the Yamuna River below the Glacier had the most even distribution with only 27.8% of all OTUs represented by the top 10. In comparison, just 2 OTUs represented 28.8% of all Delhi location OTUs.

The average richness and diversity of the groups identified by PCA (Fig 5) are shown in Fig 8. The mean richness scores ranged from 318 OTUs at the Mississippi, Ganga, Tons, and other Yamuna River locations to 361 OTUs at the Yamuna River below the Glacier. There were no statistically significant differences in the average richness between the three groups (p-value (p) = 0.633). On the other hand, there was a statistically significant difference in the mean diversity indices using Simpson's $D$ (p = 0.012) and Shannon's $H$ (p<0.001) when comparing across all groups. Subsequent post-hoc analyses with Tukey's HSD revealed that the mean Simpson diversity index of $D$ = 0.986 at the Yamuna River below the glacier was significantly higher than

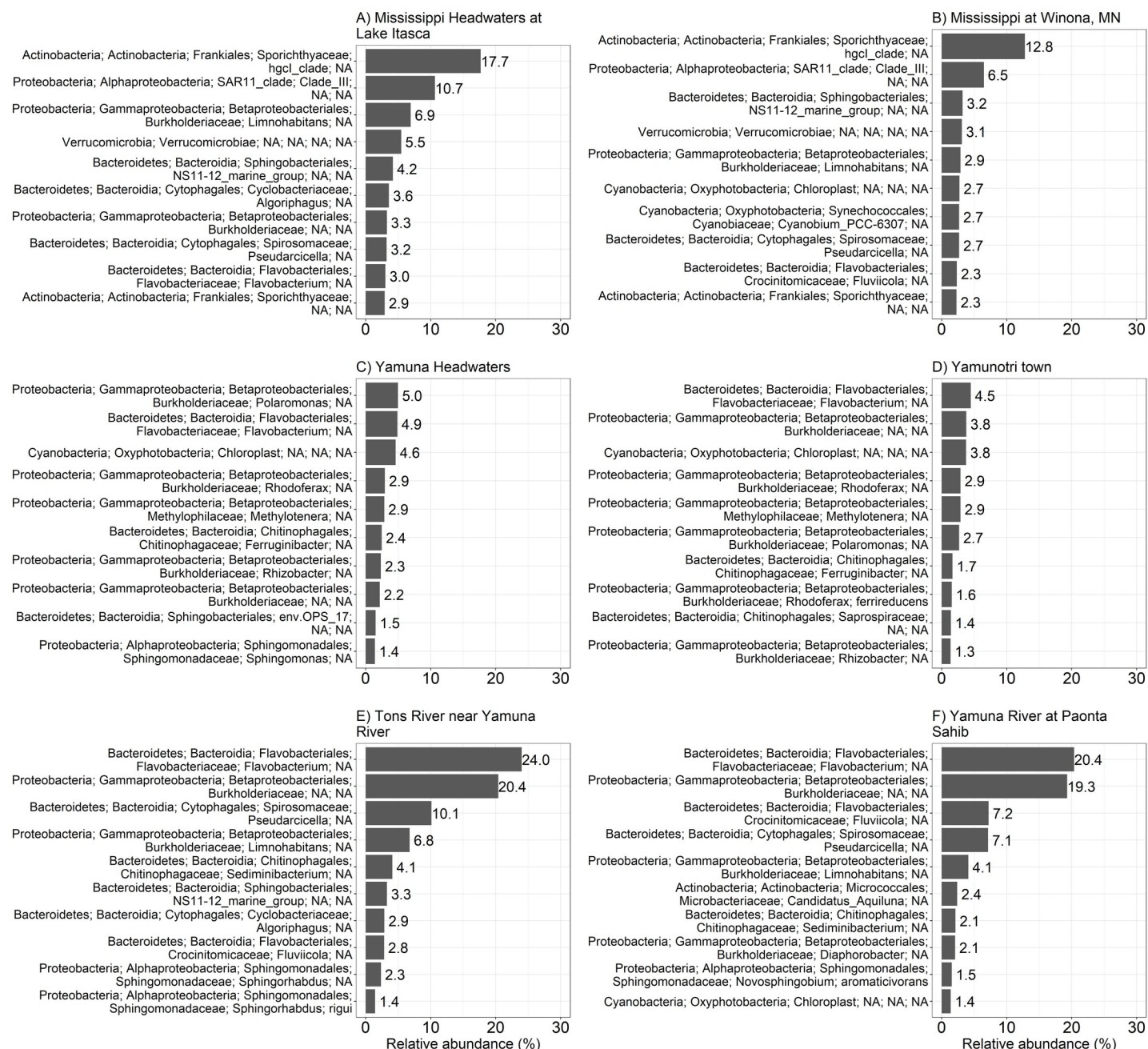

**Fig 3. Comparison of the most abundant OTUs of the Mississippi River locations and the Yamuna River locations in Uttarakhand.** Relative abundance of the 10 most abundant OTUs from (A) the Mississippi headwaters, (B) the Mississippi River at Winona, (C) Yamuna headwaters, (D) the Yamuna River at the town of Yamunotri, (E) the Tons River at its confluence with the Yamuna River, (F) Yamuna River at Paonta Sahib.

the mean $D = 0.951$ at the Delhi locations (p<0.001) and the $D = 0.943$ at Mississippi, Ganga, Tons, and other Yamuna River locations (p<0.001). The Shannon $H = 5.06$ at the Yamuna River below the Glacier was significantly higher than both the $H = 4.23$ at the Delhi locations (p<0.001) and the $H = 3.96$ at Mississippi, Ganga, Tons, and other Yamuna River locations (p<0.001). The Simpson's $D$ and Shannon's $H$ diversity indices at the Delhi locations were not

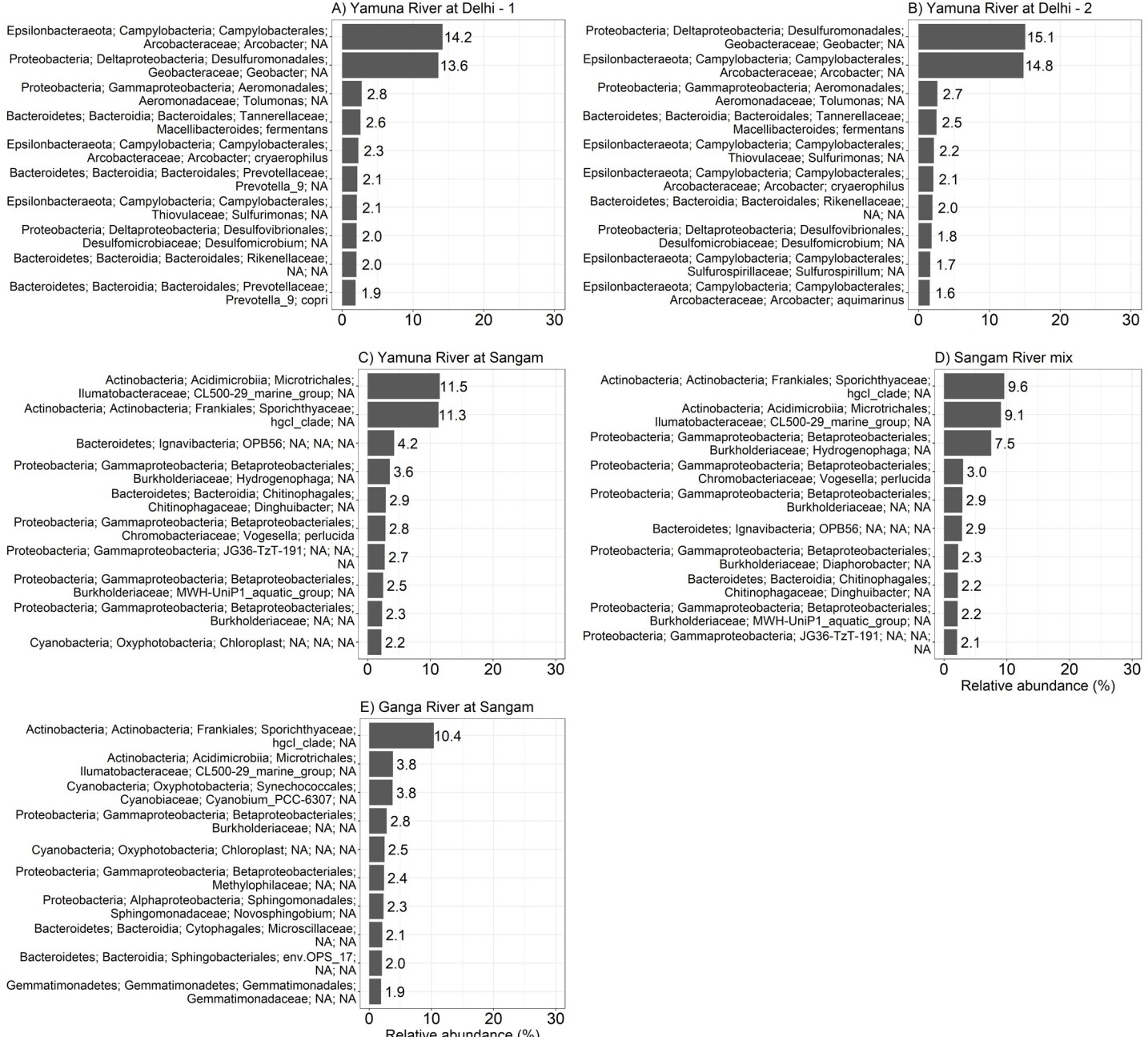

**Fig 4. Comparison of the most abundant OTUs of the Yamuna River and Ganga Rivers in Uttar Pradesh, downstream of Uttarakhand.** Relative abundance of the 10 most abundant OTUs from (A) Yamuna River in Delhi-1, (B) Yamuna River in Delhi-2, (C) Yamuna River upstream of Sangam, (D) Yamuna and Ganga River mixture at Sangam in Allahabad, (E) Ganga River upstream of Sangam.

significantly different (p = 0.5667 and p = 0.1556, respectively) from the diversity at Mississippi, Ganga, Tons, and other Yamuna River locations.

## Discussion

In this study we compared the bacterial communities in the Yamuna and Mississippi rivers of India and USA, respectively (Fig 1, sampling sites). Proteobacteria, one of the most abundant

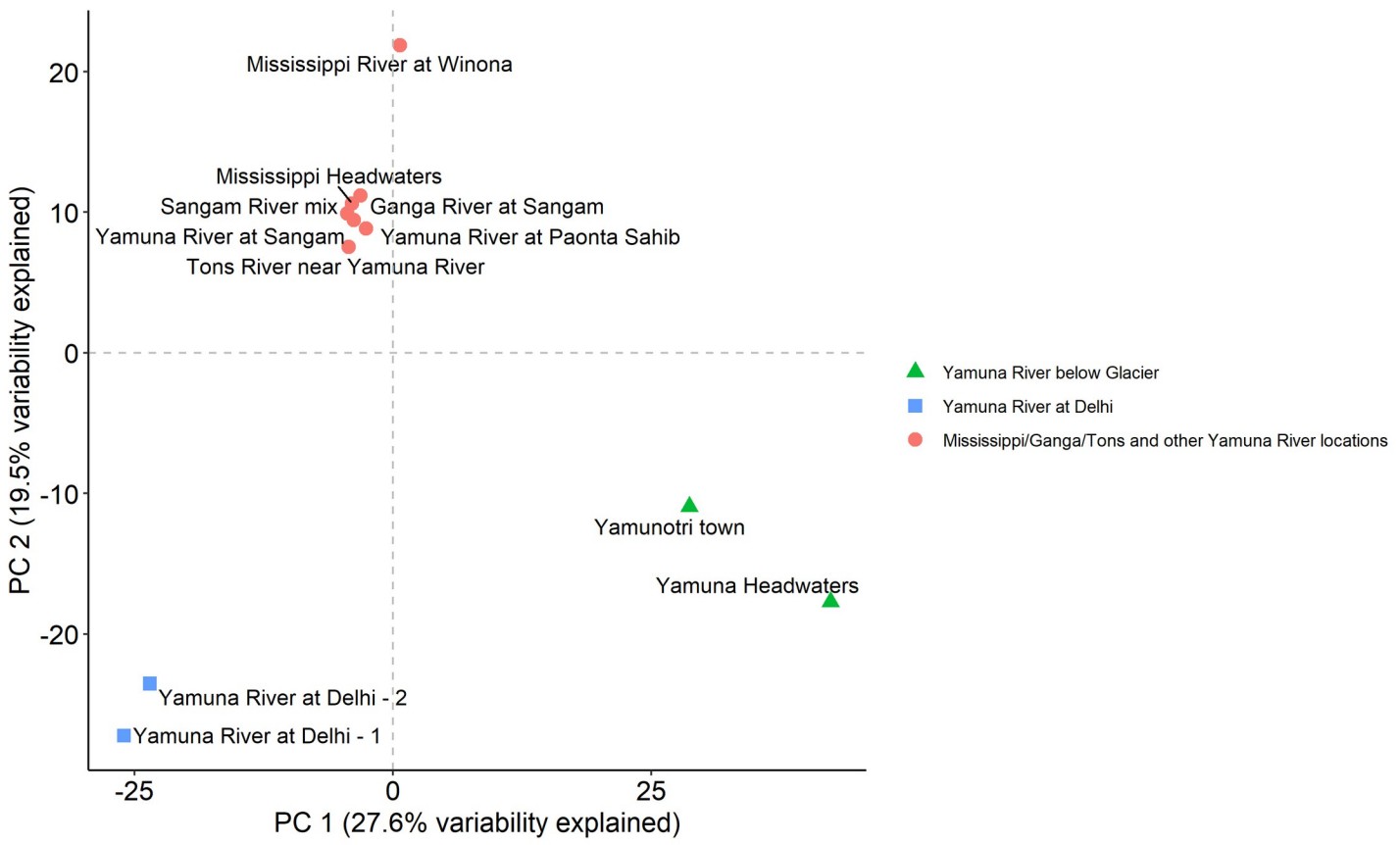

**Fig 5. Biplot of first two principal components from PCA.** Plot of the first two principal components from a PCA on OTU relative abundance at 11 locations.

and common phyla dominated river sites in the United States and in India (Figs 2 and 6) [35]. Bacteroidetes, found in abundance within the gut microbiome [36] and riverine bacterial communities was also found in abundance in all river locations. Actinobacteria, a bacterial phyla commonly found in both aquatic and terrestrial environments [37] were >5% of bacteria found in the Mississippi River and the Yamuna, Ganga, and mixture of the two rivers at Sangam (Figs 2 and 6). Consistent with our study, previous studies have shown the dominance of Proteobacteria, Bacteroidetes and Actinobacteria, amongst other phyla. For example, Cyanobacteria, Bacteroidetes, Proteobacteria, Acidobacteria, Actinobacteria and Verrucomicrobia have been shown to dominate in the upper and lower Mississippi River [13, 17, 18]. Metagenome analysis of two Yamuna River sites near Delhi by Mittal et al. demonstrated greater abundance of Proteobacteria, Bacteroidetes, Firmicutes and Actinobacteria. The Proteobacteria also dominated the bacterial community in both June (summer, pre-monsoon) and November (post-monsoon), while the numbers of Bacteroidetes, Firmicutes and Actinobacteria significantly contracted post-monsoon [16].

The *hgcl_clade* genera of the phyla Actinobacteria was the most abundant OTU at the Mississippi River in the United States as well as the Yamuna and Ganga Rivers at Sangam in India (Figs 3A, 3B, 4C–4E and 7). However, at upstream locations on the Yamuna Headwaters, Yamuna River at Yamunotri, Tons River, Yamuna River at Paonta Sahib, Yamuna River locations 1 and 2 in Delhi (Figs 3C–3F, 4A, 4B and not shown) the *hgcl_clade* was less than 1% of

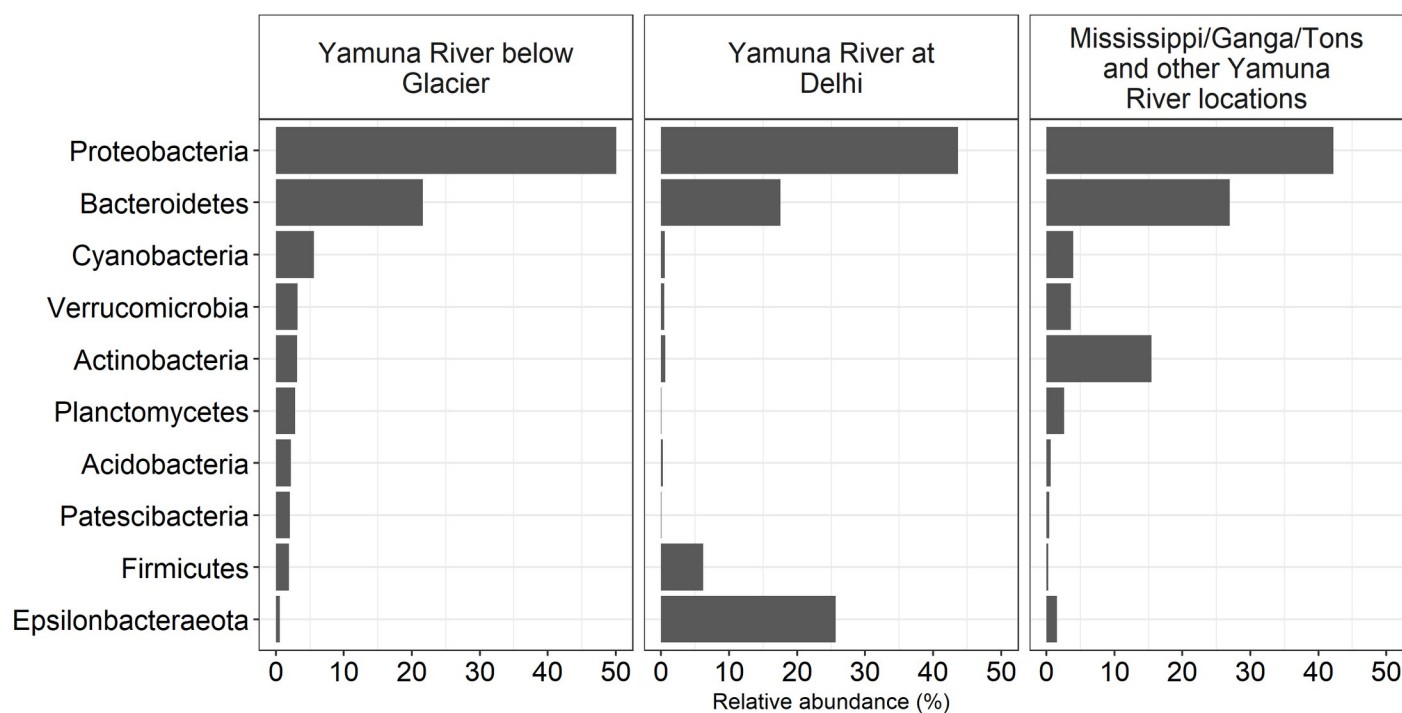

**Fig 6. Most prominent bacterial phyla in groups identified by PCA.** The most abundant bacterial phyla in the Yamuna River below Glacier, Yamuna River at Delhi and Mississippi, Ganga, Tons and other Yamuna River locations as grouped by PCA (Fig 5). Only phyla with >2% relative abundance within each group are shown.

unique OTUs (not shown) suggesting that the *hgcl_clade* population had either expanded or there was a source of high numbers of bacteria downstream of the city of Delhi.

In a study where Zufiarre et al. examined the seasonality of bacterioplankton in mountain lakes, they demonstrate the existence of 1) ice-exposed bacteria whose numbers remain stable throughout all seasons, 2) high numbers of ice-covered bacterial populations that contract during the summer thaw, and 3) bacteria normally found at relatively lower numbers that increase when other community members decrease. The *hgcl_clade* (genera) shows high abundance and little seasonal variation [38]. These Actinobacteria (phyla); Sporichthyaceae (family) and

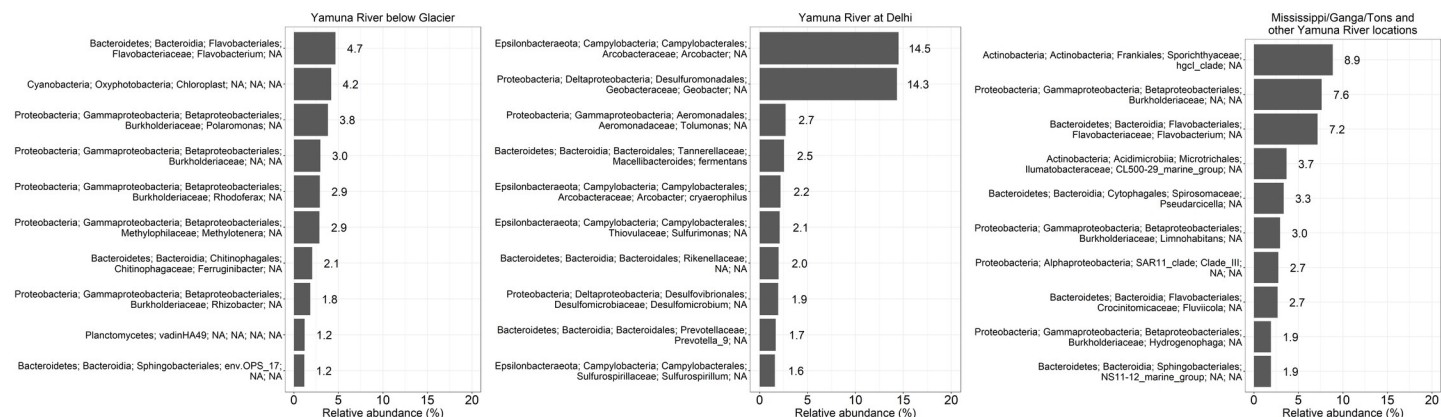

**Fig 7. Most abundant OTUs in groups identified by PCA.** The 10 most abundant OTUs in the Yamuna River below Glacier, Yamuna River at Delhi and Mississippi, Ganga, Tons and other Yamuna River locations as grouped by PCA (Fig 5).

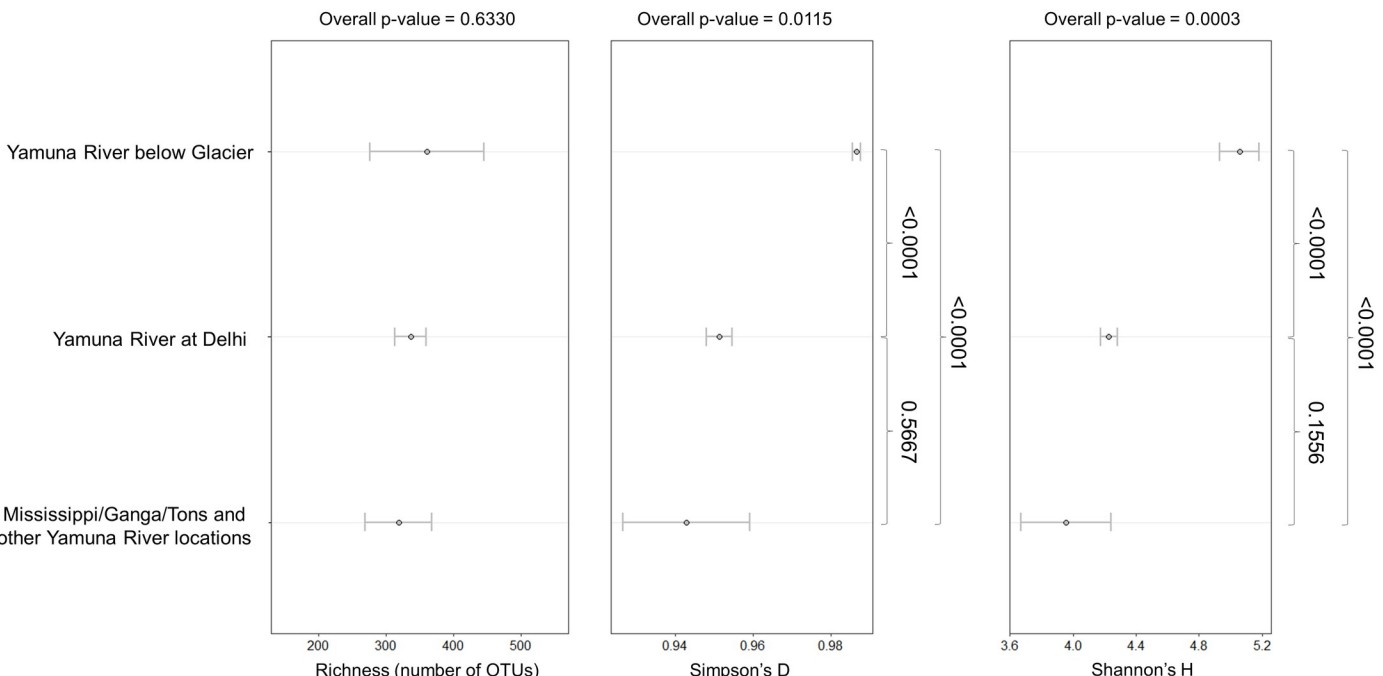

**Fig 8. Richness and diversity metrics of PCA groups.** The mean richness and diversity indices (measured by Simpson's *D* and Shannon's *H* values) were averaged across all triplicates sampled from the Yamuna River below Glacier, Yamuna River at Delhi and Mississippi, Ganga, Tons and other Yamuna River locations as grouped by PCA. Error bars represent 95% confidence intervals for the means. Where significant differences were determined by one-way ANOVA, Tukey's HSD p-values testing pairwise comparisons are shown.

*hgcl_clade* (genera) are found in terrestrial and aquatic environments [39, 40] and have adaptations that help them thrive in oligotrophic mountain lakes. Actinobacteria grow slower [40, 41] than other phyla and can store polyphosphates [42]. Genomic micro-diversification [38, 43] may explain how some *hgcl* lineages have adapted to grow in dynamic freshwater environments and are resist to seasonal changes.

*Arcobacter* and *Geobacter*, together, represented nearly 30% of all the OTUs found in the Yamuna River in Delhi (Figs 4A, 4B and 7). *Arcobacter* can be found in rivers, springs and sewage [44, 45] and have the ability to survive in different environments [46, 47]. For example, *Arcobacter* has been found in water samples from different water sources in the Kathmandu Valley, Nepal [48] as well as in the Yamuna River in Delhi (this study and [20, 49]). *Arcobacter* is also associated with humans and animals, including sheep, cattle, and chickens [45] and has been found in floodwaters and water runoff. For example, water was contaminated with *Arcobacter* after extreme precipitation and a hurricane on the South Bass Island in Ohio and North Carolina, respectively [47, 50]. *Arcobacter*, associated with fecal-contaminated water [51] and wastewater treatment plant effluent [52], may cause foodborne and waterborne infections [49]. *Campylobacter* and *Arcobacter* are members of the Campylobacteraceae family. *Campylobacter* is the cause of bacterial gastroenteritis worldwide [53, 54], but the role of *Arcobacter* in human disease is not completely clear. For example, Infections caused by *Arcobacter butzleri*, *Arcobacter cryaerophilus* and *Arcobacter skirrowii* can cause diarrhea or be asymptomatic (reviewed in [49, 55]).

*Geobacter*, one of the two main genera found in the Yamuna River at Delhi, have the ability to make electrical contacts with both extracellular electron acceptors and other organisms within their environment. They play an important role in the oxidation of various organic compounds [56].This makes them important members of any ecosystem, filling important niches within anaerobic environments. *Geobacter* found in contaminated water environments could play a role in bioremediation by, for example, the degradation of organic pollutants [6, 21, 57, 58].

Besides anthropogenic influences, differences in the climate and topography of the river locations may influence bacterial community structure. For example, Mississippi and Yamuna headwaters have average annual temperatures of 3.17˚C (Clearwater County, MN) and 19.96˚C (Uttarkashi district, Uttarakhand 134 kms away from the Yamunotri headwaters), respectively. Mississippi and Yamuna headwaters have average monthly precipitation values of 50.3 mm and 155.42 mm, respectively. The number of days with rainfall in Bemidji, Minnesota (53kms from Lake Itasca) and in the Uttarkashi district, Uttarakhand ($\geq$ 1.0 mm) are 35.54 and 120.62, respectively [59–61]. Finally, Lake Itasca and the Yamunotri town (5km from the Yamuna headwaters) are 500 meters (m) [62] and 3198 m [63], respectively, above sea level.

The *Flavobacterium* genera and Burkholderiaceae family were the most abundant OTUs (Fig 3C–3F) in the Yamuna headwaters, Yamuna River Yamunotri town, the Yamuna River at Paonta Sahib and the Tons River in the Uttarakhand, Himalayan lowlands plateau. Interestingly, the *Flavobacterium* and Burkholderiaceae river OTUs increased in relative abundance downstream of the Yamunotri town, whereas the other eight most abundant OTUs observed at the Yamuna headwaters decreased in number, disappearing from the top 10 OTUs downstream of the Yamunotri town at the Tons River and the Yamuna River at Paonta Sahib and the Yamuna River locations in Delhi and the Yamuna River upstream of Sangam (Figs 1, 3C–3F and 4). *Flavobacterium* represent a diverse set of aerobic, gram-negative mainly freshwater bacteria found throughout the world including colder environments such as the Antarctic [64]. Due to the economic importance of the fishing industry, relatively more *Flavobacterium* fish pathogens have been isolated and studied [64, 65]. Moderate thermophilic *Flavobacterium* have been isolated from thermal hot springs within the Himalayan region including thermal hot springs located in the state of Uttarakhand [66]. Consistent with this observation, the Yamuna headwaters' location in this study is directly upstream of the Yamunotri temple built around a thermal hot spring adjacent to the Yamuna River. The Yamuna River at the town of Yamunotri is located downstream of this hot spring. Although Burkholderiaceae family member OTUs were not listed in the top 10 OTUs in the Yamuna River in Delhi (Fig 4A and 4B), the OTUs were found at all locations near and at Sangam (Fig 4C–4E). The Burkholderiaceae family of bacteria can be found in many aquatic ecosystems. For example, 16S rRNA gene sequencing by Smirnova et al. [67] of two green snow samples from the eastern coast of Antarctica showed that the bacteria were mostly represented by the Burkholderiaceae (46.31%), Flavobacteriaceae (22.98%), and Pseudomonadaceae (17.66%) families.

Burkholderiaceae (family); *Polaromanus* (genus) listed within the top 10 OTUs in the Yamuna headwaters decreased in numbers such that it no longer was within the top 10 OTUs downstream of the Yamuna River at Yamunotri (Figs 3C–3F and 4). *Flavobacterium* and *Polaromanus* have been isolated from glaciers [68]. *Polaromanus*, isolated from Antarctic snow was found to grow optimally at temperatures below 18˚C [67]. Consistent with this, *Polaromanus* OTUs were found in abundance at the Yamuna headwaters (19.3˚C) and downstream at the town of Yamunotri (18.6˚C), but not at any other river site (Figs 3 and 4). Water temperatures at all other river locations were greater than 23˚C (Table 1).

Our PCA highlighted differences between the two Yamuna River locations below the glacier; the two Yamuna River locations at Delhi; and all the other locations (Fig 5). The bacterial profiles of the Yamuna River locations at Delhi PCA group demonstrated a dominance of

Firmicutes and Epsilonbacteraeota phyla (Fig 6) as well as *Arcobacter* and *Geobacter* genera (Fig 7). The dominance of a few bacteria in the Delhi river locations was also reflected in the relatively lower diversity indices (Fig 8). Conversely, the two Yamuna River locations below the Glacier PCA group had significantly higher diversity (Simpson's *D* and Shannon's *H*) as compared to the Delhi locations (both p<0.001, Fig 8) and the Mississippi, Ganga, Tons, and other Yamuna River locations (both p<0.001, Fig 8) consistent with the more even distribution of dominant phyla (Fig 6) and OTUs (Fig 7) seen below the Yamunotri glacier.

Bacterial community structure is in part a result of growth and death of bacteria seeded from a variety of sources. For example, Caporaso et al. show evidence for the existence of a persistent "seed bank" for the microbial ecosystem in the English Channel [69]. Furthermore, Staley et al. studying seasonal dynamics and bacterial reservoirs in the upper Mississippi River showed that sediment communities differed from water communities and that sediment and soil act as bacterial reservoirs [24, 25]. Furthermore, Staley et al. showed that bacterial diversity seemed highest in the soil and sediment found in the Mississippi River ecosystem [25]. Future studies examining water, soil, sediment and ice associated bacteria should clarify the source of Mississippi and Yamuna River bacterial populations.

This is the first study we know that compares the bacterial populations in the Yamuna and Mississippi Rivers of India and the USA, respectively. We show that both rivers harbour many common bacteria. For example, we demonstrate the common dominance of the Proteobacteria and Bacteroidetes phyla in the rivers of both countries. Further, the *hgcl_clade* bacterial OTU was one of the most abundant OTUs in the Mississippi River and the Ganga and Yamuna Rivers at Sangam (Fig 4) as well as in the Mississippi, Ganga, Tons and other Yamuna River locations PCA group (Fig 7). We also show some unique differences in the rivers. For example, the specific dominance of *Arcobacter* and *Geobacter* genera in the Yamuna River in Delhi.

Although the Mississippi River in the USA and the Yamuna River in India originate from distinct sources, both these rivers traverse diverse landscapes and support the local population. Both rivers face present and future challenges such as agricultural runoff, industrial discharge, and urban pollution, affecting its water quality and ecological health.

Our findings show that sites closest to the Yamunotri Glacier exhibit higher bacterial diversity, emphasizing the importance of studying and preserving the pristine quality of these water sources of immeasurable value.

## Supporting information

**S1 File.**
(PDF)

**S2 File.**
(DOCX)

## Acknowledgments

The authors want to thank Valerie Kellog for producing the maps used in Fig 1. We want to thank Neal Dale Mundahl for his constructive criticism of the manuscript.

## Author Contributions

**Conceptualization:** Osvaldo Martinez.

**Data curation:** Osvaldo Martinez, Silas R. Bergen, Jacob B. Gareis.

**Funding acquisition:** Osvaldo Martinez.

**Investigation:** Osvaldo Martinez.

**Methodology:** Osvaldo Martinez.

**Project administration:** Osvaldo Martinez.

**Software:** Silas R. Bergen.

**Supervision:** Osvaldo Martinez, Silas R. Bergen.

**Writing – original draft:** Osvaldo Martinez, Jacob B. Gareis.

**Writing – review & editing:** Osvaldo Martinez, Silas R. Bergen.

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
