## [Decision Letter · Decision Letter 0]

13 Mar 2024

PONE-D-24-07226Comparison of bacterial communities in the Yamuna River (India) and the Mississippi River (United States of America) reveals greatest diversity at the Yamuna headwaters below the Yamunotri GlacierPLOS ONE

Dear Dr. Martinez,

Thank you for submitting your manuscript to PLOS ONE. After careful consideration, we feel that it has merit but does not fully meet PLOS ONE’s publication criteria as it currently stands. Therefore, we invite you to submit a revised version of the manuscript that addresses the points raised during the review process.

We look forward to receiving your revised manuscript.

Kind regards,

Bijay Kumar Behera, Ph.D.

Academic Editor

PLOS ONE

Journal Requirements:

   "OM is a recipient of personal improvement fund (PIF), Winona State University Faculty-led Program Exploratory and Winona Foundation Grants."

4. We note that Figure 1 in your submission contain map/satellite images which may be copyrighted. All PLOS content is published under the Creative Commons Attribution License (CC BY 4.0), which means that the manuscript, images, and Supporting Information files will be freely available online, and any third party is permitted to access, download, copy, distribute, and use these materials in any way, even commercially, with proper attribution. For these reasons, we cannot publish previously copyrighted maps or satellite images created using proprietary data, such as Google software (Google Maps, Street View, and Earth). For more information, see our copyright guidelines: http://journals.plos.org/plosone/s/licenses-and-copyright.

Additional Editor Comments:

Kindly find the two reviewers comments and revise accordingly

Reviewers' comments:

Reviewer's Responses to Questions

**Comments to the Author**

1. Is the manuscript technically sound, and do the data support the conclusions?

Reviewer #1: Yes

Reviewer #2: Yes

2. Has the statistical analysis been performed appropriately and rigorously? 

Reviewer #1: Yes

Reviewer #2: N/A

3. Have the authors made all data underlying the findings in their manuscript fully available?

Reviewer #1: Yes

Reviewer #2: Yes

4. Is the manuscript presented in an intelligible fashion and written in standard English?

Reviewer #1: Yes

Reviewer #2: Yes

5. Review Comments to the Author

Reviewer #1: The manuscript, “Comparison of bacterial communities in the Yamuna River (India) and the Mississippi River (United States of America) reveals greatest diversity at the Yamuna headwaters below the Yamunotri Glacier," Manuscript Number: PONE-D-24-07226, is well written and presented.

However, the script needs some more input and corrections. Please refer to the comments below to rectify the manuscript.

1. There is no Short Title to the Manuscript. Assign a short title Before the Abstract part.

2. There are no keywords in the manuscript after the abstract. Assign keywords after the abstract part.

3. The Introduction needs significant corrections and should be written again, citing more recent works of literature. It should also contain more information on the glaciers and deltas of India and the USA.

4. Information on the Mississippi River (United States of America) is insufficient in the whole context. The river's water content quality and the natural and anthropogenic activity involved are also not cited.

5. The materials and methods part is well written but lacks the time in hours of sampling.

6. In the results section, Lines 168-169 with the table, you mentioned sampling sites as 12, but you have only given names for 11 sites. Check the Table 1.

7. In the discussion part of the manuscript, the central functioning of the prominent microbial phyla is missing. Along with the information on abundance, discussion on the functioning of the particular microbe will be more appreciated.

Reviewer #2: The title of the manuscript is not interesting and too long

Kindly modify the abstract section of the manuscript

The introduction section needs to be improved and cite some latest references

PMID: 37239442, 35596862, 32683080, 33178147, 33021988

Line no 54: Reference is missing

Line no 65: Reference is missing

Line no 129-130: Provide the individual SRA accession number

The figure quality is very poor

Need PCA/CCA analysis

6. PLOS authors have the option to publish the peer review history of their article (what does this mean?). If published, this will include your full peer review and any attached files.

Reviewer #1: **Yes: **Sushree Swati Rout

Reviewer #2: **Yes: **Ajaya Kumar Rout

---

## [Author Response · Author response to Decision Letter 0]

8 May 2024

Dear Bijay Kumar Behera,

 Thank you for considering our manuscript [PONE-D-24-07226] for publication in PLOS ONE. Please find enclosed our rebuttal to the reviewers’ comments and criticisms. Further, we have made substantial changes to the text and have included an extra 3 figures.

We have addressed the additional requirements:

1) We have checked the manuscript against PLOS ONE's style requirements

2) We have submitted a PLOS’ questionnaire on inclusivity in global research

3) With respect to the funding agencies for OM, we state that "The funders had no role in study design, data collection and analysis, decision to publish, or preparation of the manuscript." 

4) Copyright permission from the author to Fig 1

Rebuttal to Reviewer 1

 We thank reviewer 1 for their comments and criticisms. We thank reviewer 1 for commenting that our study was well written and presented.

The following is a point-by-point response to reviewer 1’s comments, criticisms and requests.

1. There is no Short Title to the Manuscript. Assign a short title Before the Abstract part.

Lines 18-19: We have added a short title before the abstract

2. There are no keywords in the manuscript after the abstract. Assign keywords after the abstract part.

Lines 45-46: We have added key words after the abstract

3. The Introduction needs significant corrections and should be written again, citing more recent works of literature. It should also contain more information on the glaciers and deltas of India and the USA.

Lines 68-84, 89-94: We thank reviewer 1 for points 3 and 4. We have made substantial changes to the manuscript, rewritten parts of the abstract, introduction, results and discussion. We have added more information on the land-use in the United States and India surrounding the river locations sampled.

4. Information on the Mississippi River (United States of America) is insufficient in the whole context. The river's water content quality and the natural and anthropogenic activity involved are also not cited.

See above

5. The materials and methods part is well written but lacks the time in hours of sampling.

Line 119: Sampling time has been added to the materials and methods.

6. In the results section, Lines 168-169 with the table, you mentioned sampling sites as 12, but you have only given names for 11 sites. Check the Table 1.

We thank the reviewer for bringing up this inconsistency. We have changed the writing to reflect the fact that there were 11 sites.

7. In the discussion part of the manuscript, the central functioning of the prominent microbial phyla is missing. Along with the information on abundance, discussion on the functioning of the particular microbe will be more appreciated.

Lines 404-411, 491-500: We thank the reviewer for this comment. We have added information on the major microbe phyla found at river sites and have expanded our description of Flavobacterium, abundant below the Yamunotri glacier.

Reviewer #2: The title of the manuscript is not interesting and too long

We thank the reviewer for constructive comments and suggestions.

Lines 1-3: We have shortened and focused the title.

Kindly modify the abstract section of the manuscript

Lines 26, 28, 33-52: We have modified the abstract. We have corrected the number of sampling sites, added a sampling site, removed diversity indices for individual sites and added the PCA analysis results including the PCA group diversity indices.

The introduction section needs to be improved and cite some latest references

PMID: 37239442, 35596862, 32683080, 33178147, 33021988

See comments to reviewer 1. We have improved the Introduction. For example, we have added land-use surrounding the Mississippi and Yamuna rivers.

We thank the reviewer very much for bringing these studies to our attention. We have added where appropriate PMID 35596862, 32683080, 33178147

Line no 54: Reference is missing.

We have added a reference at line 66

Line no 65: Reference is missing

We have added two references at line 93

Line no 129-130: Provide the individual SRA accession number

Lines 166-177: We have added all the SRX numbers to the project accession number.

The figure quality is very poor

We have uploaded high resolution figures.

Need PCA/CCA analysis

We have added a PCA analysis (Fig 5) and 3 accompanying figures (Fig 6-8) characterizing the PCA groups

---

## [Editor Report · Decision Letter 1]

16 May 2024

Comparison of Yamuna (India) and Mississippi River (United States of America) bacterial communities reveals greater diversity below the Yamunotri Glacier

PONE-D-24-07226R1

Dear Dr. Martinez,

We’re pleased to inform you that your manuscript has been judged scientifically suitable for publication and will be formally accepted for publication once it meets all outstanding technical requirements.

Kind regards,

Bijay Kumar Behera, Ph.D.

Academic Editor

PLOS ONE

Additional Editor Comments (optional):

According to the reviewers recommendations the manuscript is accepted.
---

## [Editor Report · Acceptance letter]

24 Jun 2024

PONE-D-24-07226R1 

PLOS ONE

Dear Dr. Martinez, 

I'm pleased to inform you that your manuscript has been deemed suitable for publication in PLOS ONE. Congratulations! Your manuscript is now being handed over to our production team.

Kind regards, 

on behalf of

Dr. Bijay Kumar Behera 

Academic Editor

PLOS ONE